# Temperature Dependence of Absorption and Energy Transfer Efficiency of Er^3+^/Yb^3+^/P^5+^ Co-Doped Silica Fiber Core Glasses

**DOI:** 10.3390/ma15030996

**Published:** 2022-01-27

**Authors:** Yue Cheng, Hehe Dong, Chunlei Yu, Qiubai Yang, Yan Jiao, Shikai Wang, Chongyun Shao, Lili Hu, Ye Dai

**Affiliations:** 1Key Laboratory of High Power Laser Materials, Shanghai Institute of Optics and Fine Mechanics, Chinese Academy of Sciences, Shanghai 201800, China; chengyue@siom.ac.cn (Y.C.); 18252097526@163.com (H.D.); yangqiubai@siom.ac.cn (Q.Y.); jiaoyan@siom.ac.cn (Y.J.); woshiwsk@163.com (S.W.); shaochongyun@siom.ac.cn (C.S.); 2Hangzhou Institute for Advanced Study, University of Chinese Academy of Sciences, Hangzhou 310024, China; 3Department of Physics, Shanghai University, Shanghai 200444, China

**Keywords:** Er^3+^/Yb^3+^ co-doped silica glass, temperature dependence, absorption cross-sections, energy transfer efficiency

## Abstract

A high phosphorus Er^3+^/Yb^3+^ co-doped silica (EYPS) fiber core glass was prepared using the sol-gel method combined with high-temperature sintering. The absorption spectra, emission spectra, and fluorescence decay curves were measured and compared in temperatures ranging from 300 to 480 K. Compared to 915 and 97x nm, the absorption cross-section at ~940 nm (~0.173 pm^2^) demonstrates a weaker temperature dependence. Hence, the 940 nm pump mechanism is favorable for achieving a high-power laser output at 1.5 μm. Additionally, the double-exponential fluorescence decay of Yb^3+^ ions and the emission intensity ratio of I_1018nm_/I_1534nm_ were measured to evaluate the energy transfer efficiency from Yb^3+^ ions to Er^3+^ ions. Through the external heating and active quantum defect heating methods, the emission intensity ratios of I_1018nm_/I_1534nm_ increase by 30.6% and 709.1%, respectively, from ~300 to ~480 K. The results indicate that the temperature rises significantly reduce the efficiency of the energy transfer from the Yb^3+^ to the Er^3+^ ions.

## 1. Introduction

In recent years, eye-safe 1.5 μm high-power fiber lasers have been extensively used in free-space communication, LIDAR, and range finding, which has attracted significant attention [1,2,3,4,5]. The 1.5 μm laser is mainly based on the ^4^I_13/2_ to ^4^I_15/2_ transition of the Er^3+^ ions. In silica fiber, the Er^3+^ ions have small absorption cross-sections at ~980 nm, and a serious concentration quenching effect [6,7]. The current output power record of the single-mode Er^3+^-doped silica fiber amplifier is only 107 W [8]. To increase the output power further, doping the Yb^3+^ ions can effectively improve the pump absorption coefficient [1,9]. Additionally, doping high-concentration phosphorus is also necessary for Er^3+^/Yb^3+^ co-doped silica fiber (EYDF) [10,11]. This has two advantages, namely, an improved dispersion effect of the Er^3+^ and Yb^3+^ ions [12], and a weak back energy transfer (ET) from the Er^3+^ to Yb^3+^ ions in the high phonon energy environment [13,14]. The current power record of the single-mode EYDF is 302 W, which has an optical efficiency of 56%. The gain fiber (Nufern-LMA-EYDF-25P/300-HE) was doped with ~11mol% P_2_O_5_ [10]. At present, two primary factors limit the increase in power of the EYDF: first, the significant thermal effect caused by the quantum defect (QD) [10,15,16], which reduces the pump absorption and damages the fiber coating [17,18,19]; second, the amplified spontaneous emission (ASE) and parasitic oscillations of the Yb^3+^ ions at ~1 μm, reducing the efficiency of the ET from the Yb^3+^ to the Er^3+^ ions [11]. Both of the two effects directly affect the 1.5 μm laser performance.

Generally, a ~915 nm, ~940 nm, ~97x nm LD pump, or a 1018 nm fiber laser, can be used as the pump source of the EYDF [20]. The absorption coefficients of the EYDF at ~915, ~940, ~97x, and ~1018 nm are significantly different, and are affected by the composition of the fiber core glass. The low pump absorption can achieve a smoother population inversion distribution along the EYDF [20], thereby suppressing the significant thermal effect and leading to a Yb-ASE gain [6,10,21,22]. For the high phosphorus-doped EYDF, the absorption spectrum at 940 nm is flatter than 915 nm and 97x nm [22,23]. In summary, the ~940 nm pump is a more practical scheme for achieving a high-power EYDF amplifier.

Several studies have been conducted on the doping concentration and doping ratio of the Yb^3+^ and Er^3+^ ions in silica glass [24]. The results show that when the doping concentration of the Yb^3+^ ions reaches 3–5 wt%, and the molar concentration ratio of Yb^3+^/Er^3+^ ions is 8–15, the EYDF exhibits a better laser performance [10,24,25,26]. However, the strong pump absorption of the Yb^3+^ ions inevitably causes the temperature of the glass to rise. Due to the limitation of fiber polymer coatings, the operating temperature of fiber laser is required to be lower than ~200 °C [27]. Only a few studies have compared the change of primary pump absorption cross-sections and the efficiency of the ET from the Yb^3+^ to the Er^3+^ ions with the temperature rise of the EYDF amplifier [28]. In this work, the Er^3+^/Yb^3+^ co-doped with ~15 mol% P_2_O_5_ silica glass was prepared using the sol-gel method combined with high-temperature sintering for the first time. The temperature dependence (from 300 to 480 K) of the primary absorption cross-sections (at 915, 940, 974, and 1018 nm) was calculated and analyzed. Additionally, the fluorescence decay of the Yb^3+^ ions and the emission intensity ratio of I_1018nm_/I_1534nm_ were measured to evaluate the efficiency of the ET from the Yb^3+^ to the Er^3+^ ions under two rising temperature conditions. The results provide sufficient data supporting the experimental design and modeling of the EYDF amplifier.

## 2. Experiment Details

### 2.1. Sample Preparation

The chemical composition of the high phosphorus-doped silica core glass is presented in Table 1. Both the EYPS and Yb^3+^/P^5+^-doped silica (YPS) bulk glasses were prepared using the sol-gel method, combined with high-temperature sintering, with an excellent refractive index uniformity [29]. TEOS, ErCl_3_·6H_2_O, YbCl_3_·6H_2_O, H_3_PO_4_, and C_2_H_5_OH were used as precursors. All precursors were mixed in pure water and stirred at ~30 °C to form a homogeneous rare earth-doped sol. After heating to 1000 °C, a gel powder with decomposed hydroxyl and organic matter was obtained. Transparent bulk glasses were prepared after the dry powder was melted at 1650 °C for 2 h in a vacuum state. To simulate the fiber drawing process, the bulk glasses were heat-treated in a hydrogen–oxygen flame at ~2000 °C and then rapidly cooled in air. The detailed preparation process is described in Refs. [30,31]. The thickness of the bulk glasses was polished to 1 mm. The EYPS and YPS bulk glasses contain the same concentrations of P_2_O_5_ and Yb_2_O_3_, which are 15.0mol% and 1.0 mol%, respectively. The actual concentrations of Yb_2_O_3_, Er_2_O_3_, and P_2_O_5_ were measured using a Thermo iCAP 6300 radial view inductively coupled plasma optical emission spectrometry (Thermo Fisher Scientific, Waltham, MA, USA). The decrease in phosphorus content is due to the volatilization during high-temperature sintering.

### 2.2. Characterization of the Bulk Glass

The absorption spectrum was measured using a Lambda UV-VIS-NIR spectrophotometer with a scanning step of 0.1 nm. The temperature values were set to 300, 340, 380, 420, and 480 K using the temperature controller shown in Figure 1a. As shown in Figure 1b, a high-resolution Edinburgh FLS 920 Spectrofluorometer (Edinburgh Instruments, Wales, UK). was used to measure the near-infrared emission spectrum under 915 nm pulse excitation at 300–480 K. Figure 1b shows two methods for temperature control in the vacuum chamber: ① external heating of the EYPS bulk glass using the temperature controller; ② active quantum defect heating of the EYPS bulk glass by increasing the 915 nm LD pump power, with the temperature controller turning off. The temperature was detected in real time using a Fluke Ti400 (Fluke Corporation, Everett, WA, USA) infrared thermal imager. The fluorescence decay curves of Er^3+^:^4^I_13/2_ and Yb^3+^:^2^F_5/2_ energy levels were measured at 1018 and 1534 nm under 980 nm pulse excitation, respectively. Additionally, the fluorescence decay with time evolution in the wavelength range of 1000–1700 nm under 980 nm pulse excitation was measured using an FLS920 Spectrofluorometer at ~300 K.

## 3. Results and Discussion

### 3.1. Efficient Energy Transfer of EYPS Bulk Glass

The ~15mol% P_2_O_5_ was introduced in the EYPS bulk glass, and the composition was close to that of the commercial erbium–ytterbium co-doped silica fiber (Nufern-LMA-EYDF-25P/300-HE) [10]. The high doping concentration of rare earth ions and the efficiency of the ET from the Yb^3+^ to the Er^3+^ ions were guaranteed. The fluorescence decay of the Yb^3+^ ions and emission intensity ratio of I_~1μm_/I_1534nm_ in the EYPS glass can be used to evaluate the efficiency of the ET from the Yb^3+^ to the Er^3+^ ions [2,32]. Figure 2a shows the normalized emission spectrum of the EYPS under 915 nm pump excitation. The weak emission intensity in the range of 1–1.1 μm indicates the high efficiency of the ET from the Yb^3+^ to the Er^3+^ ions. Figure 2b shows the fluorescence decay curves of the Er^3+^ and Yb^3+^ ions in the EYPS at ~300 K. The fluorescence lifetime of the Er^3+^ ions is 8.99 ms at 1534 nm. However, the decay of the Yb^3+^ is double exponential at 1018 nm. The emission intensity at 1018 nm decreases to 1/e of the initial intensity at 8.14 μs. A short fluorescence lifetime, τ_1_ (8.14 μs), and a long fluorescence lifetime, τ_2_ (~1.3 ms), were obtained. This is attributed to the existence of two types of Yb^3+^ ions in the EYPS: those closely coupled to the Er^3+^ ions and those uncoupled from the Er^3+^ ions [11,33]. The distance between the Yb^3+^ and Er^3+^ ions determines whether the two can be coupled [24], and a detailed discussion is presented in Section 3.3. In summary, the study of spectral properties based on the EYPS bulk glass has a high reference value for high-power EYDF amplifiers.

### 3.2. Temperature Dependence of the Primary Absorption Cross-Sections of the EYPS

The high-power operation of the EYDF amplifier results in a high heat load, and the fiber core temperature may reach 500 K under air-cooling conditions [10,15,16,34], which limits the power increase. Figure 3a shows the absorption cross-section curves of the Yb^3+^ ions in the EYPS bulk glass at 300, 340, 380, 420, and 480 K. The calculation formula of the absorption cross-sections were described in detail in Ref. [17]. The ratios of the primary absorption cross-section at different temperatures to that at 300 K in the EYPS bulk glass are shown in Figure 3b. As the temperature increases from 300 to 480 K, the absorption cross-sections of the EYPS bulk glass decrease by 27.2% (0.173 pm^2^→0.126 pm^2^) at 915 nm and 37.9% (0.943 pm^2^→0.586 pm^2^) at 974 nm. The absorption cross-section at 1018 nm increases by 106.9% (0.029 pm^2^→0.060 pm^2^). As we all know, the 1018 nm pump wavelength has a high quantum conversion efficiency for the 1.5 μm laser. The 106.9% increase in the absorption cross-section significantly impacts the amplification using the 1018 nm co-band pump. The higher reabsorption of the fiber core glass at 1–1.1 μm can also suppress the Yb-ASE gain. Additionally, the absorption cross-section at 940 nm is 0.173 pm^2^, which has little change with temperature. Compared with 915 and 974 nm in Figure 3b, the absorption cross-section at ~940 nm demonstrates a weak temperature dependence simultaneously, which is conducive to the stable output of the high-power EYDF amplifier.

The reduction in the absorption cross-section is caused by the temperature-dependent electron–phonon interaction at the sub-level of the rare earth ions [17,35]. The particles in the ground state are redistributed with the temperature rise, and this process conforms to the Boltzmann distribution. In Equation (1), *b_1N_* represents the fraction of the population of the *N*-th Stark sub-energy level within the ground-state manifold (where N = 1 or 2; see Figure 4) relevant to the absorption transition [35], *k_b_* is the Boltzmann constant, *T* is the temperature, and *E*_1*N*_ − *E*_11_ is the difference in energy relative to the lowest Stark energy level [17]:(1)$$b_{1N}(T)=\frac{\exp[-(E_{1N}-E_{11})/k_{b}T]}{\sum_{i=1}^{4}\exp[-(E_{1i}-E_{11})/k_{b}T]}$$

Figure 4a,b show the absorption curves of the Yb^3+^ ions in EYPS bulk glass after the multiple peaks fitting at 300 and 480 K, respectively. For the convenience of discussion, the ground state ^2^F_7/2_ and the excited state ^2^F_5/2_ of the Yb^3+^ ions are simplified into two sub-levels and three sub-levels, respectively, as shown in the inset of Figure 4a. The dotted lines represent the decomposition peaks corresponding to different Stark transitions. The absorption transitions at ~915 and 974 nm correspond to ^2^F_7/2_→^2^F_5/2_ (b_11_→b_23_) and ^2^F_7/2_→^2^F_5/2_ (b_11_→b_21_), respectively. The absorption transitions at ~940 nm correspond to ^2^F_7/2_→^2^F_5/2_ (b_11_, b_12_→b_22_), and the transitions at ~1018 nm correspond to ^2^F_7/2_→^2^F_5/2_ (b_12_→b_21_). The proportions of the integrated area of the fitted peaks corresponding to b_12_→b_22_ and b_12_→b_21_ increase, while that of the integrated area involving b_11_ decreases. According to Equation (1), as the temperature rises from 300 to 480 K, the value of b_11_ decreases by 24.0%, and the value of b_12_ increases by 31.4%. These results are consistent with the changing trend of the above-mentioned integrated area of fitted peaks. Therefore, as the temperature increases, the absorption cross-sections at ~915 and 974 nm, which are determined by the value of b_11_, significantly decrease. The absorption cross-sections at ~940 nm, which are affected by both the value of b_11_ and b_12_, are maintained. The absorption cross-sections at ~1018 nm which are determined by the value of b_12_, significantly increase.

### 3.3. Analysis of the Temperature Dependence of the Efficiency of the Energy Transfer from Yb^3+^ to Er^3+^ Ions in EYPS

The efficiency of the ET from the Yb^3+^ to the Er^3+^ ions in the EYDF directly affects the output power, the ~1 μm ASE, and the ytterbium parasitic lasing [11]. The efficiency of the ET from the Yb^3+^ to the Er^3+^ ions can be evaluated using the fluorescence decay of the Yb^3+^ ions [32]. However, the decay of the Yb^3+^ ions is double exponential in Figure 2b. Therefore, the mechanism of the double-exponential decay was first probed. Figure 5a shows the fluorescence decay curves of the EYPS bulk glass, with the time evolution in the wavelength range of 1000–1700 nm under 980 nm pulse excitation at 300 K. During the pulse duration (~28 μs), the Er^3+^ ions have exhibited a strong emission intensity at 1534 nm, due to the high efficiency of the ET from the Yb^3+^ to the Er^3+^ ions. After the pulse is stopped, the emission intensity of the Yb^3+^ ions at ~1 μm rapidly decreases to 1/e of the initial intensity within ~10 μs, and the peak emission intensity of the Er^3+^ ions at 1534 nm continues to increase. Combined with Figure 2b, τ_1_ (8.14 μs) corresponds to the ET process from the Yb^3+^ ions to the Er^3+^ ions, and τ_2_ (~1.3 ms) corresponds to the spontaneous emission (SE) process of the Yb^3+^ ions [11,32,33]. The fluorescence decay of the Yb^3+^ ions is significantly affected by the distance between the Yb^3+^ and Er^3+^ ions in the matrix [33,36,37]. Most of the Yb^3+^ and Er^3+^ ions in the EYPS are concentrated around phosphorus [38]. Therefore, two types of Yb^3+^ ions are assumed: those coupled and those uncoupled to Er^3+^. Figure 5b shows the simplified model of the coupled and isolated Yb^3+^ ions in the EYPS bulk glass. The Yb^3+^ ions closely coupled with the Er^3+^ ions show ET and SE_1_ processes simultaneously, and the emission of the Yb^3+^ ions which are not close to the Er^3+^ ions decays during the SE_2_ processes.
ET:Er3+:F15/24+Yb3+:F5/22→Er3+:F13/24+Yb3+:F7/22+Phonon
SE:Yb3+:F5/22→Yb3+:F7/22+Photon

To further study the ET and SE processes in the high-power EYDF amplifier, the fluorescence decay of the Yb^3+^ and Er^3+^ ions in the EYDF was measured under a 980 nm excitation wavelength in 300 to 480 K in Figure 6a,b. The results show that the fluorescence lifetime of the Er^3+^ ions gradually decreases because of the increased probability of the non-radiative transition under temperature rise conditions. Table 2 is obtained by fitting the fluorescence decay of the Yb^3+^ ions according to Equation (2):(2)$$I(t)=I_{0}+A_{1}\cdot e^{-t/\tau_{1}}+A_{2}\cdot e^{-t/\tau_{2}}$$

In Equation (2), τ_1_ and τ_2_ correspond to the fluorescence lifetimes of the ET and SE_2_ processes of the Yb^3+^ ions, respectively. A_1_ and A_2_ are the relative proportions of the population participating in the ET and SE_2_, respectively. As the temperature increases, the probability of the non-radiative transition of the excited electron in the Yb^3+^ ions increases, and the population participating in the ET and SE_2_ processes decreases. Table 2 shows that the values of A_1_ and A_2_ are ~86% and ~14%, respectively. As the temperature increases, the value of A_1_ decreases, while that of A_2_ increases, indicating that the proportion of the SE_2_ process increases. The value of τ_1_ decreases, owing to the increase in the ET rate, which is assisted by multi-phonons after the temperature increases. The decrease in τ_2_ and τ_Yb_ is due to the increase in the probability of non-radiative transition. In summary, as the temperature rises, the relative proportion (A_2_) of the Yb^3+^ ions involved in the SE_2_ process increases. The increase in A_2_ indicates that the relative emission intensity at ~1 μm increases, and the efficiency of the ET from the Yb^3+^ to the Er^3+^ ions decreases.

Subsequently, the emission intensity ratio of I_1018nm_/I_1534nm_ was also measured to evaluate the effect of temperature on the efficiency of the ET from the Yb^3+^ to the Er^3+^ ions. Figure 6c shows that the fluorescence spectra of the EYPS bulk glass were measured under a 50 mW 915 nm LD pump excitation at 300–480 K. With the increase in temperature caused by the external heating, the peak emission intensity of the Yb^3+^ and Er^3+^ ions in the EYPS bulk glass decreases significantly. Figure 6d shows the normalized fluorescence spectra, and the inset in Figure 6d shows an enlarged view at ~1 μm. The emission peaks broaden as the temperature rises from 300 to 480 K [1,17,35]. The emission intensity ratio of I_1018nm_/I_1534nm_ increases by 30.5%, and the result is confirmed by the increase in the A_2_ value in Table 2. The efficiency of the ET from the Yb^3+^ to the Er^3+^ ions significantly decreases as the temperature rises.

Considering the practical application of the EYDF, without changing the heat dissipation conditions, the temperature rise of the fiber core is due to the thermal effect associated with the increase in the LD pump power. The active quantum defect heating in the vacuum chamber, caused by increasing the 915 nm LD pump power, is shown in Figure 1b, with the temperature controller turning off. Similar to the above discussion, the emission intensity ratio of I_1018nm_/I_1534nm_ was measured to evaluate the effect of temperature on efficiency of the ET from the Yb^3+^ to the Er^3+^ ions. Figure 7a shows the changes of the emission intensity (I_1534nm_ and I_1018nm_) and the EYPS surface temperature with the increasing pump power. As the 915 nm LD pump power increases from 50 to 3000 mW, the surface temperature of the EYPS bulk glass increases from 305.9 to 481.1 K, caused by the significant thermal effects of LD pumping. The emission intensity I_1018nm_ continuously increases with increased pump power. However, the emission intensity I_1534nm_ reaches a maximum when the LD pump power increases to 2000 mW. The results indicate that the ET process from the Yb^3+^ to the Er^3+^ ions reaches a bottleneck. The increase in temperature and the population inversion of the Yb^3+^ and Er^3+^ ions [23,33] cooperatively limit the ET process. Figure 7b shows the normalized emission spectrum of the EYPS bulk glass under the excitation of the 915 nm LD pump. As the EYPS surface temperature rises from 305.9 to 481.1 K, the emission intensity ratio of I_1018nm_/I_1534nm_ increases by 709.1% from 0.045 to 0.3641, and the efficiency of the ET from the Yb^3+^ to the Er^3+^ ions is reduced significantly. In summary, the emission intensity ratios of I_1018nm_/I_1534nm_ in Figure 6d and Figure 7b increase by 30.6% and 709.1%, respectively. Compared with the external heating of the EYPS, the quantum defect heating not only causes a temperature increase, but results in a more significant reduction in the efficiency of the ET of the Yb^3+^ to the Er^3+^ ions. Under a high enough pump power, the ET process will be saturated. The increase in the population inversion of Yb^3+^ and Er^3+^ ions [23,33] results in a stronger ~1 μm intensity, adversely affecting the 1.5 μm laser output.

## 4. Conclusions

The Er^3+^/Yb^3+^ ion co-doped silica glasses with 15mol% P_2_O_5_ were prepared using the sol-gel combined high-temperature sintering method. The high efficiency of the ET from the Yb^3+^ to the Er^3+^ ions was guaranteed by high phosphorus doping. In this study, the temperature dependence (the range of 300 to 480 K) of the primary absorption (at 915, 940, 974, and 1018 nm) and the efficiency of the ET from the Yb^3+^ to the Er^3+^ ions in the EYPS fiber core glass were investigated. The results indicated that the absorption cross-section decreased by 27.2% (0.173 pm^2^→0.126 pm^2^) at 915 nm and 37.9% (0.943 pm^2^→0.586 pm^2^) at 974 nm, and increased by 106.9% (0.029 pm^2^→0.060 pm^2^) at 1018 nm. However, the pump absorption cross-section at ~940 nm was maintained at 0.173 pm^2^ from 300 to 480 K. The weak temperature dependence of the pump absorption at ~940 nm is conducive to the stable high-power output of EYDF amplifiers.

Additionally, the double-exponential fluorescence decay of the Yb^3+^ ions was observed, corresponding to the ET and SE_2_ processes. This is attributed to the existence of two types of Yb^3+^ ions in the EYPS glass: those coupled and those uncoupled to the Er^3+^ ions. As the temperature increased from 300 to 480 K because of the external heating, the population of isolated Yb^3+^ ions participating in the SE_2_ process increased, and the emission intensity ratio of I_1018nm_/I_1534nm_ increased by 30.5%. The reduction in the efficiency of the ET from the Yb^3+^ to the Er^3+^ ions was confirmed as the temperature rose. Considering the practical application of the EYDF, the quantum defect heating scheme was adopted by increasing the LD pump power. The surface temperature of the EYPS bulk glass increased from ~306 to ~481 K. The emission intensity ratio of I_1018nm_/I_1534nm_ increased by 709.1%, which is primarily attributed to the increase in the population inversion of the Yb^3+^ and Er^3+^ ions. The saturated ET process resulted in a strong ~1 μm emission intensity, and the efficiency of the ET from the Yb^3+^ to the Er^3+^ ions decreased significantly.

## Figures and Tables

**Figure 1 materials-15-00996-f001:**
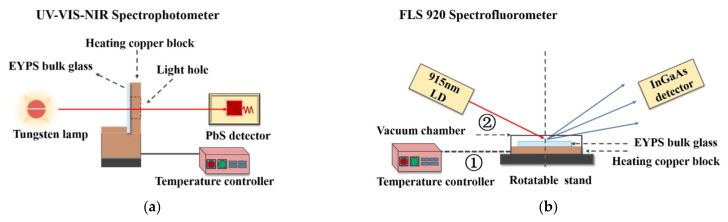
Front view of (**a**) the temperature-controllable absorption test device; front view of (**b**) the fluorescence test device, using two methods for temperature control: ① external heating of the EYPS bulk glass using the temperature controller; ② active quantum defect heating of the EYPS bulk glass by increasing the 915 nm LD pump power, with the temperature controller turning off.

**Figure 2 materials-15-00996-f002:**
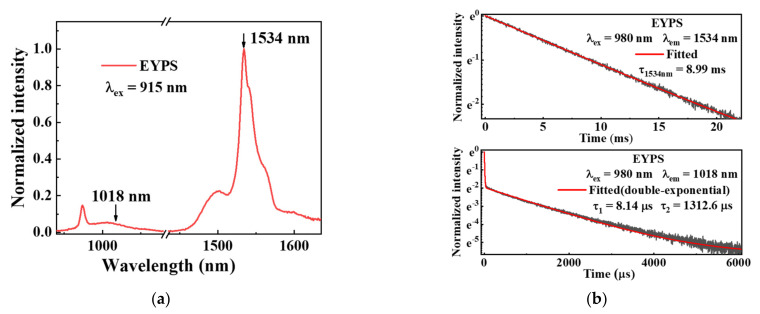
(**a**) Fluorescence spectrum of the EYPS at ~300 K; (**b**) fluorescence decay curves of Yb^3+^ and Er^3+^ ions in the EYPS at ~300 K.

**Figure 3 materials-15-00996-f003:**
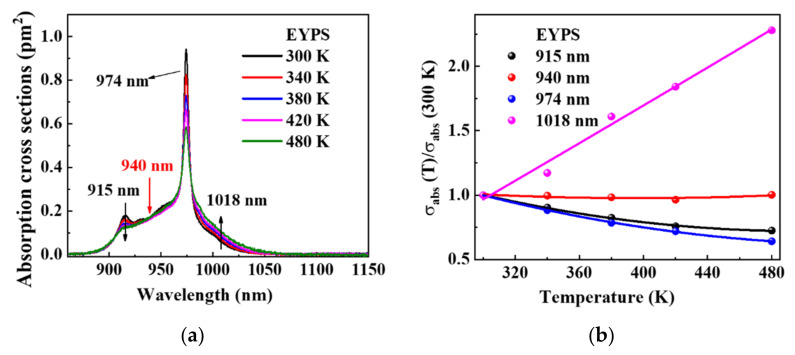
(**a**) Absorption cross-section spectra of Yb^3+^ ion in EYPS bulk glass in the temperature range from 300 to 480 K; (**b**) ratios of the absorption cross-section at different temperatures to that at 300 K for the primary pump wavelength of the EYPS bulk glass (the curves are obtained by fitting).

**Figure 4 materials-15-00996-f004:**
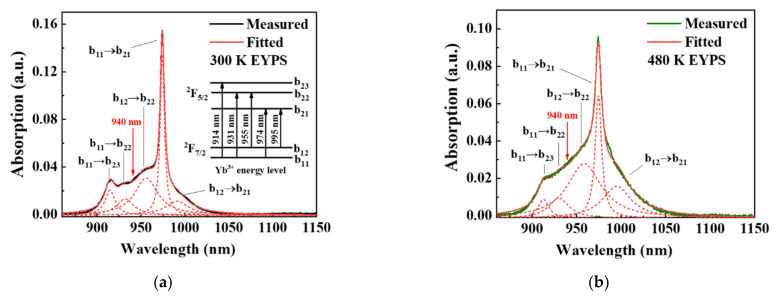
Absorption curves of Yb^3+^ ions in EYPS bulk glass are processed by Lorentz fitting at (**a**) 300 K and (**b**) 480 K. The dashed curves show the multiple decomposition peaks corresponding to different Stark transitions, and the inset is a simplified Yb^3+^ ion energy level.

**Figure 5 materials-15-00996-f005:**
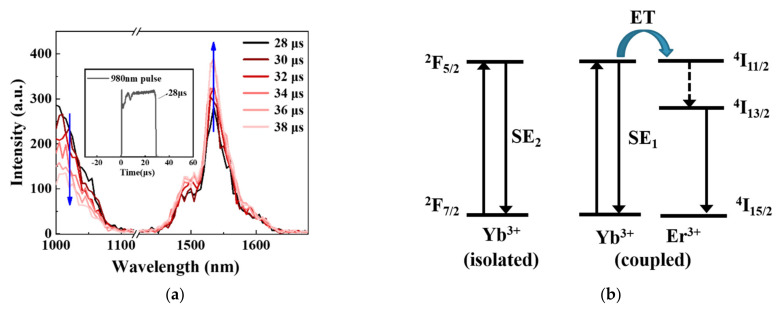
(**a**) Time−resolved emission spectra of the EYPS bulk glass under 980 nm pulse excitation at 300 K, and the inset shows the duration of the 980 nm pulse; (**b**) simplified modeling of the ET and SE processes of coupled and isolated Yb^3+^ ions in the EYPS bulk glass.

**Figure 6 materials-15-00996-f006:**
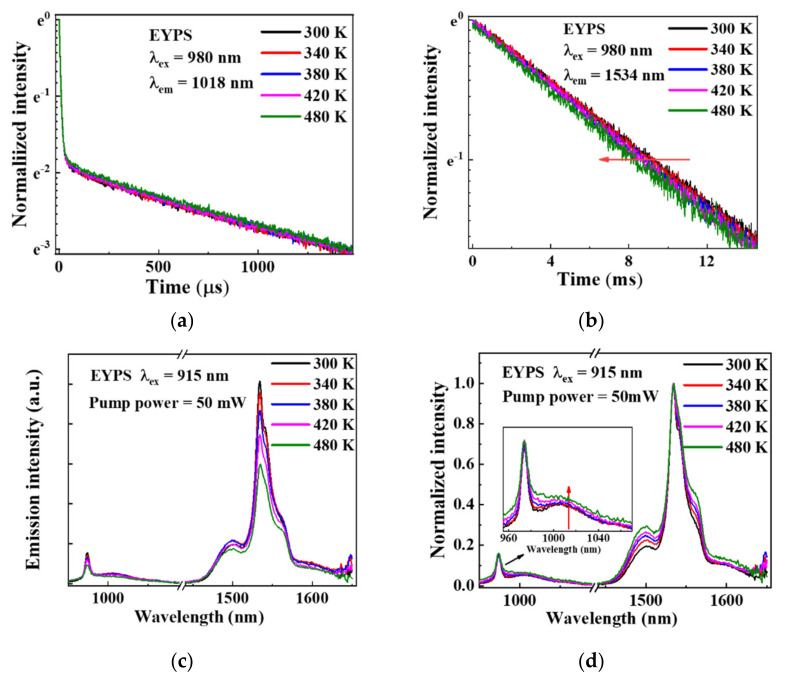
(**a**) Double−exponential fluorescence decay curves of Yb^3+^ ions and (**b**) fluorescence decay curves of Er^3+^ ions in the EYPS in 300−480 K; (**c**) fluorescence spectra of the EYPS and (**d**) its normalized spectra in 300−480 K. The inset is an enlarged view at ~1 μm, and the external heating caused the temperature rise.

**Figure 7 materials-15-00996-f007:**
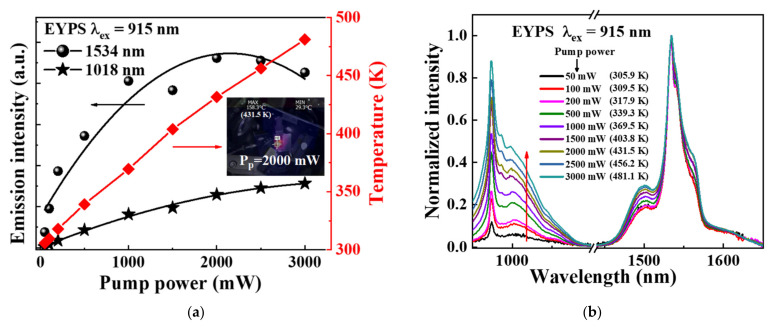
Under the 915 nm LD pump excitation in the power range of 50–3000 mW, (**a**) the change in the emission intensity (I_1534nm_ and I_1018nm_) and surface temperature of the EYPS with increasing pump power. The inset shows the surface temperature of the EYPS in real time when the pump power is increased to 2000 mw; (**b**) the normalized emission spectra of the EYPS bulk glass. The active quantum defect heating caused the temperature rise.

**Table 1 materials-15-00996-t001:** Mean compositions of the EYPS and YPS bulk glasses (mol%).

Samples	Theoretical Composition				Actual Composition			
	Er_2_O_3_	Yb_2_O_3_	P_2_O_5_	SiO_2_	Er_2_O_3_	Yb_2_O_3_	P_2_O_5_	SiO_2_
EYPS	0.1	1.0	15	83.9	0.09	0.86	12.45	86.60
YPS	-	1.0	15	84.0	-	0.83	11.08	88.09

**Table 2 materials-15-00996-t002:** Fitted results of fluorescence decay curves of Yb^3+^ and Er^3+^ ions in the EYPS and YPS bulk glasses from 300 to 480 K.

Temperature	EYPS					YPS
	A_1_	τ_1_ (μs)	A_2_	τ_2_ (μs)	τ_Er_ (ms)	τ_Yb_ (μs)
300 K	0.864	8.20	0.136	1333.76	9.12	1793.46
340 K	0.862	8.14	0.138	1312.62	8.92	1754.72
380 K	0.859	7.91	0.141	1306.87	8.76	1736.54
420 K	0.859	7.90	0.141	1299.42	8.76	1719.04
480 K	0.854	7.56	0.146	1279.25	8.14	1686.01

## Data Availability

The data presented in this study are available on request from the corresponding author.

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
