# Peer review of "Temperature Dependence of Absorption and Energy Transfer Efficiency of Er3+/Yb3+/P5+ Co-Doped Silica Fiber Core Glasses"

_materials, 2022, doi:10.3390/ma15030996_

Round 1

Reviewer 1 Report

The authors of this article discussed the temperature dependence of absorption and energy transfer efficiency of Er3+/Yb3+/P5+ co-doped silica fiber core glasses. To the best of my knowledge, the presented results are original and well within the scope of the Materials journal. The paper is well-written and contains very useful data and analysis. I then recommend its publication after the authors had the opportunity to respond the comments listed below (MA means mandatory, MI means minor)

MA1. The investigated bulk glasses have been developed by the sol-gel technique that is not the most (or the sole) used technique to such design rare-earth doped optical fibers. Throughout the paper, this point may be more discussed and in particular, it will be interested to discuss if the obtained results can be extended to glasses manufactured through different approaches.

MA2. These results are interesting but are mainly used to understand what will appear at the fiber level under high pumping of the core glass. Then it seems to me very important to discuss in the paper if the obtained results on the bulk material can be extrapolated to the fiber core material. My question relies on the possible impact of the drawing process on the temperature dependence of the spectroscopic properties of the rare-earth ions. Could you comment please? Similarly for high power EYDFA, double cladding fiber structures are used, could the device architecture influence the temperature dependence of the core material too?

MI1 - In the context part of your article, you report on the use of high-power amplifiers for free space communication applications. You maybe interested by the following article that directly discuss the temperature effects on EYDFAs with regards to space needs, more at the device level than the material level (DOI: 10.1109/TNS.2021.3069016). You could also mention that phosphorus doping if very interesting for enhancing the amplifying properties of the fiber renders it very radiation sensitive implying for those application to add as a codopant Cerium for example.

MI2. I understand that the sol-gel manufacturing process selected to manufacture the sample is described in your ref 28, however I would like to recommend to add the key details in your article in order that the paper is more self-consistent.

MI3. On p3, we state that 2 types of Yb3+ ions are present in the EYPS, here again it will be simpler to add 1-2 sentences to describe those different species in addition to the given references. I guess they correspond to the later discussed "isolated" and "coupled" species?

MI4 p5line 160 “these results” instead of “this results”

MI5. Figure 7a), in the caption you should discuss the inset (picture) that is quite hard to really analyze. Same thing for Fig.5a and the 980nm pulse inset, it should be introduced in the figure caption too.

MI6. You could also better explain how the temperature range investigated has been selected for your study

Thanks for this very interesting work,

best regards,

Reviewer 2 Report

The manuscript entitled "Temperature dependence of absorption and energy transfer efficiency of Er3+/Yb3+/P5+ co-doped silica fiber core glasses" presents the detailed and well-designed study on the optical properties of the Er/Yb co-doped P2O5-SiO2 glass. 

The authors investigated the temperature dependence of the different Yb3+ absorption bands and the energy transfer efficiency from Yb3+ to Er3+ ions using the state-of-the-art spectroscopic techniques. It was found that PL excitation around 940 nm is the most preferable for applications in high-power amplifiers.

Additionally, the double-exponential fluorescence decay of Yb3+ ions was observed and attributed to the existence of two types of Yb3+ ions in the studied glass.

I recommend to accept the manuscript for publication in Materials after the slight update of the Section 2.

1) Authors should specify in more details the preparation technique for the studied glass samples. Since the scope of the Materials journal directed to the research and development of new materials detailed description of the experimental part is crucial.

2) Authors should specify the model of the device used for the ICP-OES as well as the detection limits and sample preparation technique.

3) Authors should described how the "absorption cross-section" values were determined.

Round 2

Reviewer 1 Report

Dear Authors,
Many thanks for considering my comments. I recommend acceptance of the article as it is for the Materials journal
Best regards,